# Examining kinesin processivity within a general gating framework

Johan OL Andreasson[1†‡], Bojan Milic[2†§], Geng-Yuan Chen[3], Nicholas R Guydosh[4¶], William O Hancock[3], Steven M Block[2,5*]

[1]Department of Physics, Stanford University, Stanford, United States; [2]Department of Biology, Stanford University, Stanford, United States; [3]Department of Biomedical Engineering, Pennsylvania State University, University Park, United States; [4]Biophysics Program, Stanford University, Stanford, United States; [5]Department of Applied Physics, Stanford University, Stanford, United States

*For correspondence: sblock@stanford.edu

[†]These authors contributed equally to this work

Present address: [‡]Department of Genetics, Stanford University School of Medicine, Stanford University, Stanford, United States; [§]Biophysics Program, Stanford University, Stanford, United States; [¶]Department of Molecular Biology and Genetics, Johns Hopkins University School of Medicine, Howard Hughes Medical Institute, Johns Hopkins University, Baltimore, United States

**Competing interests:** The authors declare that no competing interests exist.

**Abstract** Kinesin-1 is a dimeric motor that transports cargo along microtubules, taking 8.2-nm steps in a hand-over-hand fashion. The ATP hydrolysis cycles of its two heads are maintained out of phase by a series of gating mechanisms, which lead to processive runs averaging ~1 μm. A key structural element for inter-head coordination is the neck linker (NL), which connects the heads to the stalk. To examine the role of the NL in regulating stepping, we investigated NL mutants of various lengths using single-molecule optical trapping and bulk fluorescence approaches in the context of a general framework for gating. Our results show that, although inter-head tension enhances motor velocity, it is crucial neither for inter-head coordination nor for rapid rear-head release. Furthermore, cysteine-light mutants do not produce wild-type motility under load. We conclude that kinesin-1 is primarily front-head gated, and that NL length is tuned to enhance unidirectional processivity and velocity.

## Introduction

Kinesin-1, hereafter referred to simply as kinesin, is an ATP-driven, dimeric motor protein that facilitates the unidirectional transport of intracellular cargo along microtubule (MT) filaments (*Vale et al., 1985*; *Howard et al., 1989*; *Block et al., 1990*; *Hackney, 1995*). Each kinesin dimer is composed of a pair of identical catalytic motor domains, or heads, which are connected to a common, coiled-coil stalk by a ~14-amino-acid-long sequence known as the neck linker (NL) (*Kozielski et al., 1997*). Kinesin translocates towards the plus-ends of MTs (*Svoboda et al., 1993*) via an asymmetric hand-over-hand mechanism (*Asbury et al., 2003*; *Yildiz et al., 2004*), hydrolyzing one molecule of ATP (*Hua et al., 1997*; *Schnitzer and Block, 1997*; *Coy et al., 1999*) for each 8.2-nm step (*Svoboda et al., 1993*) the motor takes. The biochemical states associated with each kinesin head during stepping are coupled to mechanical transitions in an overall mechanochemical cycle: biochemical events modulate the affinities of heads to the MT and influence mechanical transitions, and mechanical states, in turn, influence the rates of biochemical processes (*Block, 2007*).

During the mechanochemical cycle, each kinesin head transitions between one or more states that are strongly bound to the MT (the ATP-containing state, and also the no-nucleotide state), and one or more states that are weakly bound to the MT (the ADP-containing state) (*Block, 2007*). A simplified version of this cycle (*Figure 1A*) may arbitrarily be taken to begin with the one-head-bound (1-HB), ATP-waiting state [α], where the nucleotide-free front head is strongly bound to the MT while the rear, ADP-bound tethered head remains unbound (*Hackney, 1994*; *Asenjo and Sosa, 2009*; *Guydosh and Block, 2009*; *Toprak et al., 2009*). Following ATP binding to the MT-bound head, the NL of its motor domain undergoes a structural reconfiguration and forms a β-sheet with the head, in a process termed

**eLife digest** In cells, molecules are moved from one location to another by motor proteins. Kinesins are a large family of such motors that transport their cargos along long filaments known as microtubules. Most kinesin molecules are formed from two identical protein chains. Each chain has a motor region at one end (called the head) that can attach to microtubules. The other end of each protein chain wraps around its partner to form a common stalk region (called the tail) that links to the cargo being carried.

The two kinesin heads are connected to the tail via a 'neck linker' region, and they advance along the microtubule in strict alternation, similar to the way our legs move when walking. During each step, the front head remains tightly associated with the filament as the trailing head releases itself, advances beyond the front head, and reattaches to become the new leading head. The two heads need to coordinate their activities, so that at any given time, they're not at the same stage in the process. For example, if both heads remained bound to the microtubule at the same time, the motor would not be able to advance. If they both released, the motor would fall off the filament and diffuse away. However, the process by which the heads coordinate is not fully understood, and different models for how this process works have been proposed.

Now, Andreasson, Milic et al. have examined the role played by the neck linker in coordinating the motor's movement using a technique known as 'optical trapping'. The experiments involved attaching microscopic beads to the motor proteins, which serve as markers that can be tracked. The beads can also be used to exert controlled forces on the kinesin molecules, to see how they respond to different loads.

Andreasson, Milic et al. extended the length of neck linker by inserting extra amino acids (which are the building blocks of proteins) into this region of the protein. It was found that kinesins can still walk even when each neck linker was extended by up to six additional amino acids. However, introducing even a single amino acid into the linker relaxed the normal tension that exists between the heads when these are both bound to the filament. This resulted in slowed speeds, shorter distances of travel, and less ability to sustain loads. The experimental results suggest that the length of the neck linker in naturally occurring kinesins may be optimized to support maximum movement. Based on their data, Andreasson, Milic et al. propose a general framework for understanding the communication that needs to take place between the heads in order to walk in a coordinated manner. Further work is required to understand if motor proteins other than kinesins can also be understood with this same framework.

NL docking (*Rice et al., 1999*; *Schnitzer et al., 2000*; *Rosenfeld et al., 2001*; *Asenjo et al., 2006*; *Tomishige et al., 2006*; *Khalil et al., 2008*; *Sindelar and Downing, 2010*; *Clancy et al., 2011*). NL docking shifts the position of the tethered head towards the MT plus-end, beyond the position of the bound head [$\beta_1$, $\beta_2$]. Recent work has suggested that NL docking may occur in two stages: first, ATP binding induces a load-dependent mechanical transition, leading to partial NL docking [$\beta_1$], whereas subsequent ATP hydrolysis completes the docking and enables the tethered head to bind the MT [$\beta_2$] (*Milic et al., 2014*). Once the bound head hydrolyzes ATP, fully docks its NL, and the tethered head binds the MT, the motor enters a mechanically strained, two-heads-bound (2-HB) state [$\gamma$] (*Rice et al., 1999*; *Rosenfeld et al., 2003*; *Block, 2007*; *Yildiz et al., 2008*; *Gennerich and Vale, 2009*; *Clancy et al., 2011*). Finally, rear-head release (*Klumpp et al., 2004*) completes the step and returns the motor to its initial 1-HB waiting state [$\alpha$], primed to begin the cycle anew, after having translocated forward by one step (8.2 nm) along the MT. The processivity of kinesin—evinced by the ability of a single dimer to undergo >100 consecutive stepping cycles before dissociation—relies upon a tight coordination of the biochemical and mechanical events, collectively known as *gating mechanisms* (*Block, 2007*).

We note that the term 'gating' encompasses any mechanism where the state of one head influences its partner in such a way as to ensure that the mechanochemical cycles of the heads are maintained out of phase, leading to alternate-head stepping. This term includes, but is not limited to, mechanisms that modulate detachment rates, alter nucleotide affinities, and affect ATP hydrolysis (*Block et al., 1990*; *Hackney, 1994*; *Vale et al., 1996*; *Hua et al., 1997*; *Schnitzer and Block, 1997*; *Hancock and Howard, 1999*; *Rosenfeld et al., 2003*; *Block, 2007*).

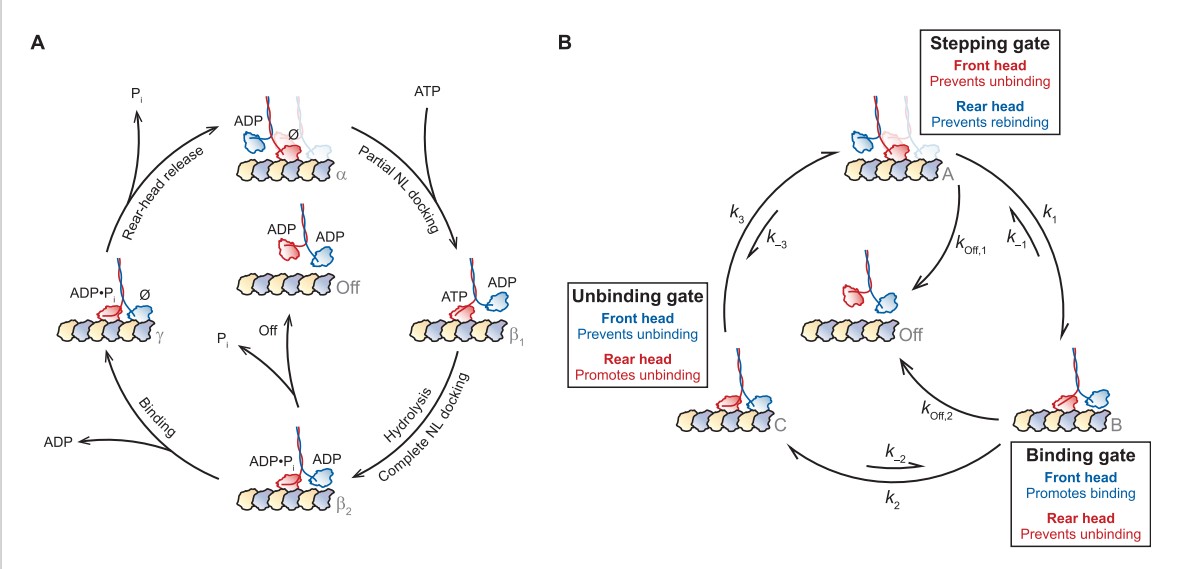

**Figure 1**. A general gating framework based on mechanical states of dimeric motors. (**A**) The kinesin mechanochemical cycle. Kinesin starts from the one-head-bound (1-HB) ATP-waiting state [α], characterized by a strongly bound, nucleotide-free (Ø) front head (red) and an unbound, ADP-containing tethered head (blue). ATP binding induces a force-dependent transition involving partial NL docking, shifting the tethered head past the bound head [β₁]. ATP hydrolysis completes NL docking and facilitates tethered-head binding [β₂]. At this point, kinesin may access a dissociated state [Off], induced by premature phosphate release from the bound head, leading to dimer detachment. However, if the tethered head reaches the forward MT binding site and completes the step before the bound head can dissociate, kinesin enters the two-heads-bound (2-HB) state [γ]. Rear-head release returns the dimer to the ATP-waiting state [α], having moved forward by 8.2 nm. (**B**) A simplified general gating framework, based on the cycle in (**A**). Stepping, binding, and unbinding gates are shown with associated rate constants between each of the three gated states, [A], [B], and [C]. The cycle begins at the 1-HB ATP-waiting state [A], where the stepping gate promotes processivity by inhibiting rear-head (blue) rebinding and premature bound-head (red) release. Following a force-dependent step that shifts the tethered head past the bound head [B], the binding gate promotes binding of the tethered head at the forward MT binding site while inhibiting release of the bound head. Also shown is a competing dissociated state [Off], arising from premature release of the bound head from the 1-HB state, accessible from either [A] or [B]. Tethered-head binding leads to the 2-HB state [C], where the unbinding gate promotes rear-head release while inhibiting front-head release, returning the motor to the start of the cycle [A].

To establish a general framework for gating in the kinesin cycle, we begin by recognizing that the kinesin dimer transitions through a fixed series of 1-HB [A, B] and 2-HB [C] states during each cycle, where the unbound (tethered) head of a 1-HB motor may be positioned either predominantly behind [A] or in front of [B], the MT-bound head (*Figure 1B*). Since dissociation from the 2-HB state [Off] necessarily requires passage through a 1-HB intermediate, processive cycling can be distilled into the following set of three gating properties which, taken all together, promote forward stepping while suppressing dissociation:

  i. The MT-bound head must remain attached in any 1-HB state.
 ii. When the motor is in either a 2-HB state or in a 1-HB state where the tethered head is behind the MT-bound head, the unbinding of the rear head should be promoted (or maintained, if already unbound).
iii. When the motor is in either a 2-HB state or in a 1-HB state where the tethered head is in front of the MT-bound head, the binding of the front head should be promoted (or maintained, if already bound).

Taken together, these principles lead to the mechanical gating framework presented in *Figure 1B*. Starting from a 1-HB state, with the tethered head positioned behind the MT-bound head [A], unidirectional processivity necessitates a *stepping gate* that stabilizes binding by the front head while inhibiting any rebinding by the rear head. Following a structural transition that shifts the tethered head beyond the bound head [B], a *binding gate* acts to promote tethered-head binding at the forward site, while preventing premature unbinding of the rear head, which would otherwise lead to dissociation [Off]. Once the tethered head binds successfully, and the motor achieves the 2-HB

state [C], an *unbinding gate* is required to retain the front head on the MT, while promoting the release of the rear head. Finally, rear-head release completes the cycle, returning the motor to its initial state [A], but advanced by one step. We note that this abstraction is agnostic with respect to the biochemical state associated with each mechanical transition. However, precisely because these biochemical states are unspecified, this general framework (*Figure 1B*) can be applied not only to kinesin but also to other processive, two-headed motors that may couple mechanical and biochemical states differently (*Block, 2007*; *Gennerich and Vale, 2009*; *Kull and Endow, 2013*; *Cleary et al., 2014*).

Here, we used the gating framework to guide an investigation of the role of the NL domain in determining kinesin processivity, which remains incompletely understood, despite considerable research (*Hackney et al., 2003*; *Block, 2007*; *Yildiz et al., 2008*; *Shastry and Hancock, 2010*, *2011*; *Clancy et al., 2011*). Because intramolecular forces within the kinesin dimer are thought to be transmitted through the NL of each head, the NL domain is well situated to play a role in gating. Consistent with this, extending the NL by mutation has previously been shown to affect kinesin processivity, velocity, and stepping behavior (*Hackney et al., 2003*; *Block, 2007*; *Yildiz et al., 2008*; *Shastry and Hancock, 2010*, *2011*; *Clancy et al., 2011*). In principle, perturbing the properties of the NL by inserting additional amino acids (AA) at the junction of the NL domain and the coiled-coil stalk could affect some, or all, of the gates in *Figure 1B*. When kinesin is in its 1-HB state [A, B], the NL might serve to suppress rear-head rebinding in the ATP-waiting state [A], as part of the stepping gate, or to promote tethered-head binding (following NL docking) in the bound head [B], as part of the binding gate. When kinesin is in a 2-HB state [C], the NL can transmit inter-head tension, affecting the unbinding gate (*Rosenfeld et al., 2003*; *Guydosh and Block, 2006*; *Block, 2007*; *Yildiz et al., 2008*; *Hariharan and Hancock, 2009*; *Shastry and Hancock, 2010*, *2011*; *Clancy et al., 2011*). The unbinding gate may act through (i) *front-head gating*, where biochemical events on the front head are suppressed until the rear head has a chance to detach (candidate mechanisms include the suppression of ATP hydrolysis in the front head, reduced ATP binding arising from inter-head tension, or reduced ATP binding caused by the rearward-pointing configuration of the NL), or through (ii) *rear-head gating*, where the rear-head release rate is accelerated by the inter-head tension, or (iii) some combination of the two (*Block, 2007*). Because inter-head tension is directly controlled by the length of the NL domain, an understanding of the relationship between NL length and the unbinding gate is central to evaluating tension-based gating models.

The predominant mechanism for gating used by kinesin has been the subject of some controversy. Published models have invoked both versions of rear-head gating (*Hancock and Howard, 1999*; *Crevel et al., 2004*; *Schief et al., 2004*; *Yildiz et al., 2008*) and front-head gating (*Rosenfeld et al., 2003*; *Klumpp et al., 2004*; *Guydosh and Block, 2006*, *2009*; *Toprak et al., 2009*; *Clancy et al., 2011*). The conversation about kinesin gating has heretofore focused on how inter-head coordination might be achieved from the 2-HB state, that is, from the unbinding gate. Comparatively little attention has been paid to gating at other points of the kinesin cycle, specifically at the stepping and binding gates (*Figure 1B*), and quantitative measures of the competing rates responsible for gating have been notably lacking. Here, we examine the quantitative contribution of each of the three gates to maintaining unidirectional processivity by assessing the effects of NL length on kinesin motility, and attempt to reframe the discussion of gating from a debate about front- vs rear-head gating at the 2-HB state to a more unified view that admits to gating at multiple states of the mechanochemical cycle, both 1-HB and 2-HB. Towards that end, we examined a series of truncated *Drosophila* kinesin constructs (DmK) containing NLs that were extended incrementally, from 1 to 6 AA (DmK-1AA to DmK-6AA), using a combination of single-molecule optical trapping and bulk fluorescence approaches.

## Results

### Kinesin maintains a stable, one-head-bound ATP-waiting state with as many as three extra amino acids in its neck linker

Our first experiments were designed to probe the influence of NL length on the *stepping gate*, which might contribute to unidirectional processivity by suppressing the rebinding of the ADP-bound tethered head while the motor is in the ATP-waiting state [A] (*Figure 1B*). To investigate the influence of NL length on this rebinding, we performed a series of half-site ADP release experiments

using mantADP, a fluorescent ADP analog, on a set of truncated constructs with extended NL domains. Because free kinesin heads in solution have a high affinity for ADP, whereas MT-bound heads exhibit a substantially lower affinity, the release of mantADP from kinesin heads serves as a proxy for the binding of free heads to MTs (*Hackney, 1988*, *1994*, *2002*; *Gilbert et al., 1995*; *Hackney et al., 2003*; *Clancy et al., 2011*). Binding was assayed by pre-incubating motors in mantADP and subsequently monitoring the MT-stimulated decay of fluorescence upon the introduction of MTs via stopped-flow. Wild-type (WT) *Drosophila* kinesin (DmK-WT), as well as constructs with NLs extended by up to 3 AA, displayed a 50% reduction in fluorescence following the addition of MTs (*Figure 2A*), consistent with being in a 1-HB state, where only one of the two heads is bound to the MT and releases mantADP. By contrast, constructs with NL inserts consisting of 4, 5, or 6 AA exhibited progressively greater drops in fluorescence upon the addition of MTs (*Figure 2A*). These decreases correspond to 16% of DmK-4AA, 38% of DmK-5AA, and 65% of DmK-6AA motors releasing mantADP from both heads, respectively. In principle, the increased mantADP release could arise from (i) the additional NL length facilitating transient rear-head rebinding events, (ii) more motors adopting a stable 2-HB ATP-waiting state, or (iii) some combination of both. These findings are consistent with previous work, which has also reported a ~50% fluorescence decrease for WT kinesin, and a nearly 100% fluorescence decrease for constructs carrying a 6-AA NL insert (*Hackney, 1994*, *2002*; *Hackney et al., 2003*; *Clancy et al., 2011*). Additional mantADP exchange experiments—where the frequency of transient rear-head interactions with the MT was assessed based on the effective rate of mantADP exchange at the rear head—revealed an order-of-magnitude increase relative to WT kinesin once the NL domain was extended by 4 or more AA (*Figure 2B*). The low mantADP exchange rates observed for our series of constructs are consistent with previous reports (*Hackney, 2002*; *Hackney et al., 2003*). Taken all together, these results indicate that kinesin can maintain a stable, 1-HB ATP-waiting state with up to 3 additional AA in its NL, but that extending the NL by 4 AA or more leads to an abrupt increase in rebinding of the rear-head to the MT.

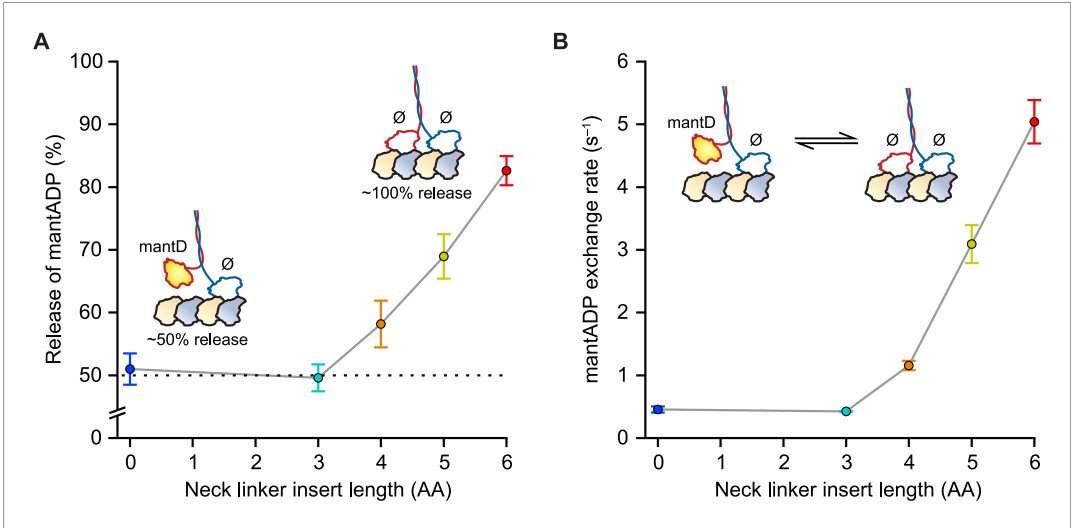

**Figure 2**. Kinesin with as many as 3 AA inserted in the NL maintains a 1-HB ATP-waiting state. (**A**) Half-site mantADP release measurements as a function of NL insert length (mean ± SE; *N* = 3). Upon MT binding, both DmK-WT and DmK-3AA, pre-incubated with mantADP (mantD), lose ~50% of their initial fluorescence. The fluorescence loss exceeds 50% for constructs containing NL inserts longer than 3 AA. Inset, A 50% loss of fluorescence corresponds to dimers binding to MTs in a 1-HB state, whereas a 100% fluorescence decrease is consistent with the release of all bound mantADP (mantD) upon MT binding. (**B**) MantADP exchange by the tethered head as a function of NL insert length (mean ± SE; *N* = 3), measured by rapidly diluting mantADP·kinesin·MT complexes into nucleotide-free buffer via stopped flow. The cartoon depicts the measured reaction. The exchange rate increased significantly for constructs with NL inserts of 4 AA or more. In the insets (**A** and **B**), white shading indicates non-fluorescent, nucleotide-free heads (∅); yellow indicates fluorescent, mantADP-bound heads.

## Run lengths are dramatically reduced in kinesin constructs with one or more additional amino acids in the neck linker

After kinesin undertakes its force-producing mechanical step [B], the *binding gate* may promote front-head binding to the forward MT binding site while keeping the rear head bound to the MT (*Figure 1B*). The overall distance kinesin travels along the MT before dissociating—the run length—is governed by the probability of dissociation during each stepping cycle, which is itself determined, to a first approximation, by a competition between the binding of the tethered front head and the premature release of the MT-bound rear head (*Milic et al., 2014*). To determine whether the rate of front-head binding at the binding gate depends upon NL length, we performed run-length measurements as a function of load for constructs with extended NLs.

Extending the NL by just a single AA decreased the unloaded run length by a factor of ~3 relative to the WT run length. Further increments in the NL length (up to 6 AA) yielded only minor decreases in run length relative to DmK-1AA (*Figure 3*). Under unloaded conditions, DmK-6AA was capable of taking several dozen consecutive forward steps, on average, before dissociating, indicating that kinesin retains significant processivity with as many as 6 AA introduced into its NL. In the presence of forces applied either against (hindering loads) or along (assisting loads) the direction of kinesin motility (*Figure 3*, inset), the differences in processivity among constructs with differing NL lengths were substantially more pronounced under hindering forces than under assisting forces. Although the unloaded run lengths for DmK-WT and DmK-1AA differed by a factor of 3, the differences in run

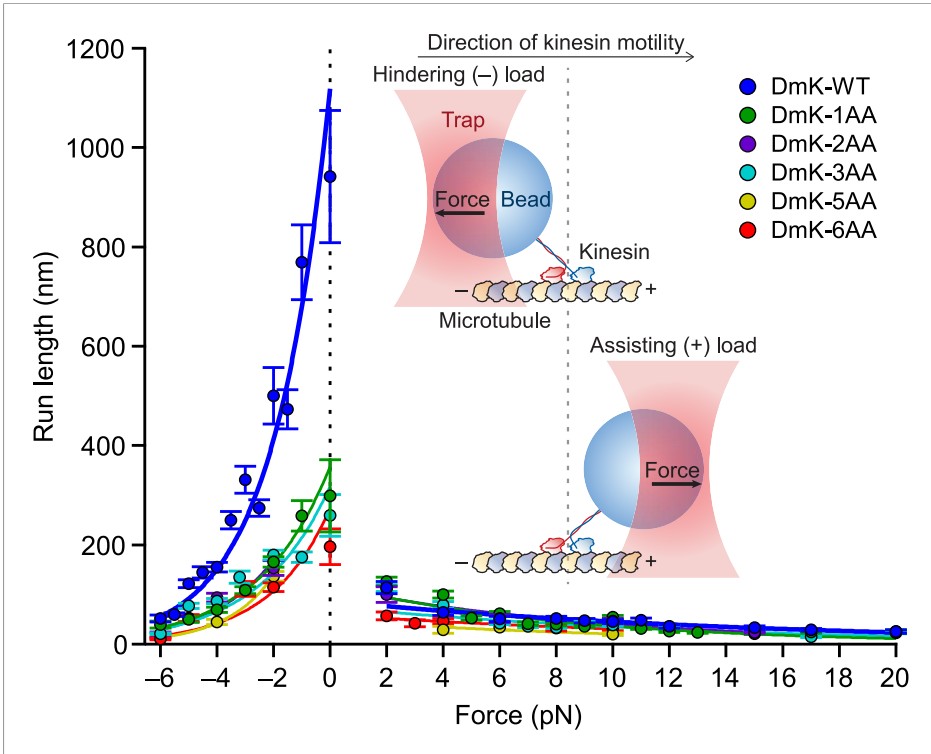

**Figure 3**. Extending the NL by a single AA compromises processivity. Mean run lengths as a function of applied force (mean ± SE; N = 49–818) for the constructs studied, acquired with an optical force clamp in the presence of 2 mM ATP (solid circles; color-coded according to the legend). DmK-WT (*Milic et al., 2014*) and hindering load data sets for DmK-3AA (*Andreasson et al., 2015*) are reproduced from our previous work. For all constructs, mean run lengths exhibited significant asymmetry, depending upon the direction of load. To obtain the unloaded run length ($L_0$) and characteristic distance parameter ($\delta_L$) for each construct, run length (L) data for hindering (−6 to 0 pN) and assisting loads (+2 to +20 pN) were separately fit to exponentials (solid lines; color-coded according to the legend) of the form $L(F) = L_0 \exp[-|F|\delta_L/k_B T]$, where F is the force applied by the optical trap and $k_B T$ is Boltzmann's constant times the absolute temperature; parameter values are in *Table 1*. Inset cartoon, a graphical representation of the experimental geometry of the single-molecule assay (not to scale).

length between these constructs under assisting loads were comparatively minor. A gradual, exponential decrease in run length was observed for all constructs under increasing hindering loads, but the application of even a small assisting force (+2 pN) abruptly reduced the run length for both the WT and NL-mutant constructs. Based on exponential fits to the mean run lengths as a function of applied load, the characteristic distance parameter for the force dependence of run length, $\delta_L$, was nearly an order of magnitude greater under hindering-load conditions than under assisting-load conditions for all constructs (*Table 1*). These results show that the highly asymmetric force dependence of DmK run lengths previously reported for the WT motor (*Milic et al., 2014*) is also exhibited by kinesin constructs with NLs extended up to 6 AA. Interestingly, although processivity was found to depend upon the NL length, particularly under hindering-load and unloaded conditions, the distance parameters for both hindering and assisting loads were nearly independent of NL length (*Table 1*).

## Added phosphate proportionally enhances run lengths of DmK-WT and DmK-6AA

Previous work has shown that under assisting loads, added phosphate (P$_i$) nearly doubles the run length of DmK-WT (*Figure 4A*), indicating that the probability of dissociation at the binding gate—and therefore run length—is determined by a competition between the rate of P$_i$ release from the bound head and the rate of productive tethered-head binding (*Milic et al., 2014*). To determine the extent to which the NL length affects this competition, we examined the influence of 100 mM potassium phosphate on the processivity of DmK-6AA, under moderate assisting loads (+4 pN). Under saturating levels of ATP, the addition of P$_i$ increased the DmK-6AA run length relative to that in its absence (*Figure 4A*). However, no statistically significant changes in processivity were detected at low ATP concentrations, nor under conditions where ATP was replaced with the slowly hydrolyzable analog, ATPγS, nor in the presence of 100 mM potassium chloride (used as a control for ionic strength effects) (*Figure 4*). Although run lengths under otherwise identical assay conditions were systematically lower for DmK-6AA than for DmK-WT, those differences disappeared when the data were normalized relative to the baseline run lengths (no added P$_i$) for each motor (*Figure 4B*).

## The velocity of kinesin constructs with extended neck linkers can be rescued by assisting loads

Once the tethered head binds the MT to generate the 2-HB state [C], the *unbinding gate* may ensure unidirectional processivity by inhibiting front-head unbinding while promoting rear-head release (*Figure 1B*). As discussed previously, the unbinding gate might consist of a rear-head gating mechanism, where the inter-head tension enhances the rate of rear-head release. If detachment of the trailing head is accelerated by inter-head tension, then extending the NL would be expected to relieve the tension and decrease the rear-head release rate, thereby reducing the overall rate at which kinesin proceeds around the reaction cycle. To explore the effect of inter-head tension on rear-head gating,

---

**Table 1**. Parameters from exponential fits to the run length data of *Figure 3*

| Construct | $L_{0-}$ (nm)* | $\delta_{L-}$ (nm)* | $L_{0+}$ (nm)* | $\delta_{L+}$ (nm)* |
|---|---|---|---|---|
| DmK-WT† | 1120 ± 60 | 2.0 ± 0.1 | 87 ± 6 | 0.27 ± 0.03 |
| DmK-1AA | 360 ± 30 | 1.6 ± 0.1 | 120 ± 7 | 0.48 ± 0.02 |
| DmK-2AA | 410 ± 60 | 1.8 ± 0.2 | 115 ± 17 | 0.42 ± 0.05 |
| DmK-3AA | 320 ± 20 | 1.6 ± 0.1 | 76 ± 4 | 0.31 ± 0.02 |
| DmK-5AA | 440 ± 60 | 2.4 ± 0.2 | 48 ± 14 | 0.35 ± 0.14 |
| DmK-6AA | 270 ± 30 | 1.9 ± 0.1 | 59 ± 8 | 0.27 ± 0.10 |

$L_{0-}$, unloaded run length for hindering loads (−6 to 0 pN); $\delta_{L-}$, distance parameter for hindering loads (−6 to 0 pN); $L_{0+}$, unloaded run length for assisting loads (+2 to +20 pN); $\delta_{L+}$, distance parameter for assisting loads (+2 to +20 pN).
*Parameter values correspond to mean ± SE.
†Values from *Milic et al. (2014)*.

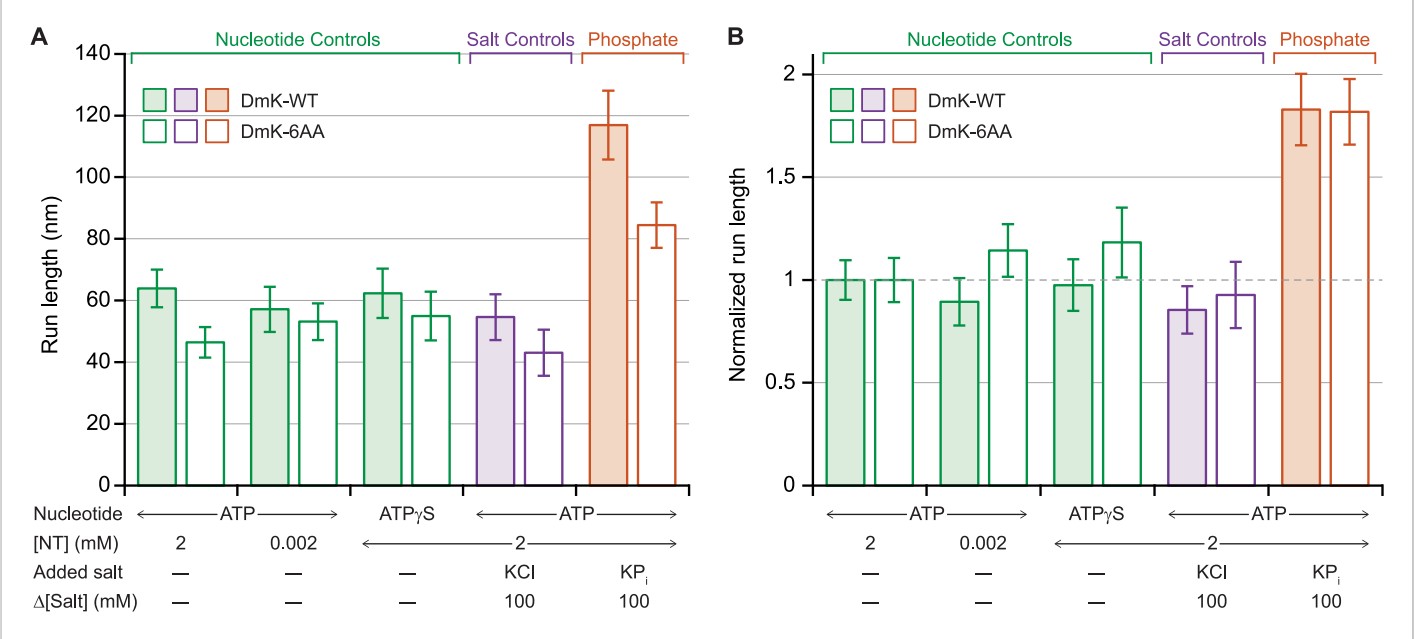

**Figure 4**. Added phosphate enhances the processivity of DmK-WT and DmK-6AA. (**A**) Mean run lengths (mean ± SE; $N = 84–210$) under moderate assisting load (+4 pN), in the presence of nucleotide analogs (green), 100 mM potassium chloride (KCl; purple), or 100 mM potassium phosphate (KP$_i$; orange). Run lengths for DmK-WT (shaded bars, data from *Milic et al., 2014*) are shown paired with DmK-6AA data (unshaded bars). Decreasing the ATP concentration, replacing ATP by ATPγS, or adding KCl elicited no statistically significant change in run length relative to the baseline run length for saturating ATP (2 mM) in the absence of added salt. The mean run length increased significantly in the presence of phosphate for both DmK-WT ($p < 10^{-4}$; *t*-test) and DmK-6AA ($p < 10^{-4}$; *t*-test). (**B**) Run-length data from (**A**), normalized to the baseline run length value for each construct.

we performed single-molecule measurements of the velocity of our kinesin constructs as a function of load.

Both WT and mutant constructs exhibited a force–velocity relationship that is highly asymmetric with respect to the direction of the applied load (*Figure 5*), consistent with previous findings (*Block et al., 2003*; *Block, 2007*). Whereas the insertion of a single AA in the NL elicited a drop in velocity across all loads, additional NL insertions produced no further changes in velocity: indeed, *Drosophila* kinesin stepped at significant rates with as many as 6 additional AA in its NL. In contrast to DmK-WT, which is not sped up by assisting loads, the reduction in unloaded velocity produced by NL extension could be recovered by applying larger assisting loads (∼20 pN). These data are consistent with the explanation that inter-head tension, which can promote rear-head release, is effectively abolished by extending the NL past its WT length, but that the reduced tension can be restored by applying assisting load, which places differential stress on the rear head. However, the finding that assisting load fails to appreciably increase the WT velocity suggests that the rate of rear-head release by WT kinesin must be substantially higher than the rate-limiting step(s) of the mechanochemical cycle. If the rate of rear-head release were instead rate-limiting, then any acceleration of that step would have manifested as an increase in motor velocity.

## The force dependence of kinesin mutants with extended neck linkers can be accounted for by a minimal 3-state model

To gain a quantitative understanding of the velocities of mutant constructs, and to gain additional insight into how inter-head tension affects kinesin gating, the force–velocity data (*Figure 5*) were fit to a minimal, 3-state model of the mechanochemical cycle (*Figure 5*, inset). The first transition in this cycle consists of the force-producing mechanical step, [1] → [2], which is modeled by a force-dependent rate, $k_1 = k_1^0 \exp[F_{trap}\delta_1/k_B T]$ where $k_1^0$ is the unloaded rate constant, $F_{trap}$ is the applied force, $\delta_1$ is a characteristic distance parameter, and $k_B T$ is Boltzmann's constant times the absolute temperature. Because kinesin is in a 1-HB ATP-waiting state [1] prior to this transition, the inter-head

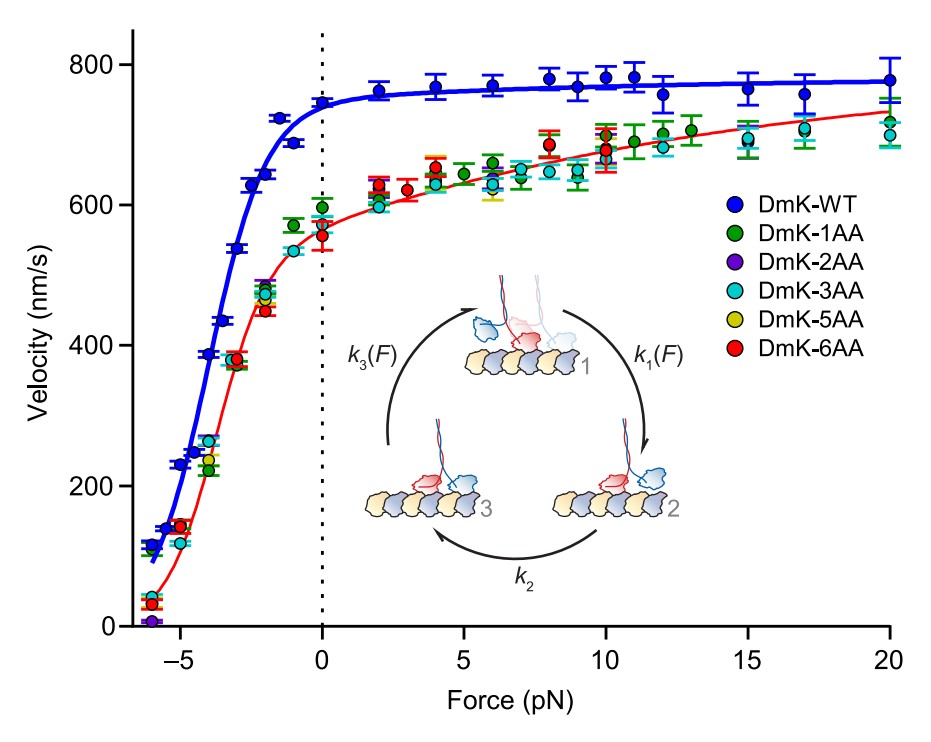

**Figure 5**. Assisting load can rescue the velocity of mutants with extended NLs. Velocity (mean ± SE; $N$ = 49–818) as a function of force for constructs (solid circles; color-coded according to the legend). Data were collected under the same conditions as *Figure 3*. DmK-WT velocity was not affected by assisting loads, but the velocities of all mutant constructs could be increased by larger assisting loads. Solid lines show the global fit to a minimal 3-state model (*inset*) for WT (blue) and mutant (red) constructs, with parameters in *Table 2*. Data sets for both DmK-WT and DmK-3AA under hindering loads are from (*Andreasson et al., 2015*). Force–velocity data are compared to other mutant constructs in *Figure 7*.

tension, $F_i$, has no effect on $k_1$. The following transition, [2] → [3], consists of ATP hydrolysis and related events—including the completion of NL docking by the MT-bound head, front-head binding to the MT, and ADP release by the new front head—that induce kinesin to enter the strained, 2-HB state [3]. Because this transition corresponds to the completion of the key biochemical steps in the cycle, the associated rate constant, $k_2$, is modeled as being independent of load. From the 2-HB state [3], the final transition of the cycle consists of rear-head release (which is also load-dependent) and is modeled as $k_3 = k_3^0 \exp[(F_{\text{trap}} + F_i)\delta_3/k_B T]$, where $k_3^0$ is the unloaded rate constant and $\delta_3$ is the associated distance parameter. The expression for $k_3$ is a function of both $F_{\text{trap}}$ and $F_i$, and accounts for the roles of both internal and external tension in enhancing the dissociation of the rear head. A previously described analytical method was adopted to derive the expression for velocity as a function of force based on this model (*Chemla et al., 2008*). This model represents a minimal extension of previously published models (*Schnitzer et al., 2000*; *Block et al., 2003*).

The 3-state model was fit globally to our data (*Figure 5*), and the resulting parameters and uncertainties are given in *Table 2*. Because the NL-insert mutants exhibited indistinguishable force-dependent velocities, we took the inter-head tension in these constructs to be negligible compared to the WT ($F_{i,\text{mutant}} \approx 0$ pN). With this assumption, we determined the inter-head tension in WT kinesin ($F_{i,\text{WT}}$) to be 26 ± 3 pN in the 2-HB state, a value that agrees well with molecular dynamics simulations, which predicted that it would lie in the range of 15–35 pN (*Hariharan and Hancock, 2009*). The model fit also supplies an estimate of the unloaded rate of rear-head release, 260 ± 10 s$^{-1}$. With the derived values for $F_{i,\text{WT}}$ and the load-dependence of rear-head unbinding, $\delta_3$, the expression for $k_3$ shows that inter-head tension alone is capable of increasing the rate of rear-head release by an order of magnitude (*Table 2*). Moreover, the modeled rate of ATP hydrolysis (plus the other primarily

Table 2. Kinetic parameters from a global fit of the 3-state model to the force–velocity data of *Figure 5*

| Parameter | Parameter description | Value* |
|---|---|---|
| $k_1^0(F)$ | Rate of ATP binding†; mechanical step | 4900 ± 300 s$^{-1}$ |
| $\delta_{1,WT}$ | Distance parameter (wild-type) | 4.6 ± 0.1 nm |
| $\delta_{1,mutant}$ | Distance parameter (mutant constructs) | 4.0 ± 0.1 nm |
| $k_2$ | Rate of ATP hydrolysis; biochemical events | 95 ± 1 s$^{-1}$ |
| $k_3^0(F)$ | Rate of rear-head release | 260 ± 10 s$^{-1}$ |
| $\delta_3$ | Distance parameter (rear-head release) | 0.35 ± 0.02 nm |
| $F_{i,WT}$ | Inter-head tension (wild-type) | 26 ± 3 pN |
| $F_{i,mutant}$ | Inter-head tension (mutant constructs) | 0 pN (fixed) |

*Parameter values correspond to mean ± SE.
†Rate for saturating ATP conditions (2 mM).

biochemical events, above), 95 ± 1 s$^{-1}$, is consistent with previous estimates of the hydrolysis rate from both biochemical and single-molecule studies (*Ma and Taylor, 1997*; *Gilbert et al., 1998*; *Schnitzer et al., 2000*; *Farrell et al., 2002*). Under all practical loads, hydrolysis is substantially slower than rear-head release.

We note that any force-dependent transition in the kinesin cycle must be associated with a corresponding movement of a head, or other subdomain, of the motor. Previous work has shown that the main force-dependent transition occurs from a 1-HB state after ATP binding (*Schnitzer et al., 2000*; *Block et al., 2003*). The velocity decrease under hindering loads is associated with this transition, which we model as $k_1$, with a correspondingly large distance parameter ($\delta_1 \geq 4$ nm; *Table 2*), equivalent to nearly half the size of the kinesin step. This transition becomes rate limiting at moderate hindering loads, and will therefore dominate any other possible force-dependent steps at high loads. Under assisting loads, $k_1$ is exceedingly fast (*Table 2*) and does not appreciably contribute to the completion time of the overall cycle—and therefore to the velocity—in this regime. However, for the mutant constructs, we are able to detect a second force-dependent contribution that is manifested as increased velocity under assisting loads (*Figure 5*). This force dependence, $\delta_3$, is necessarily linked to some motion of the motor, and the load dependence of the associated rate, $k_3$, could be attributed either to rear head release when kinesin is in the 2-HB state or to binding of the forward-positioned tethered head of a 1-HB motor. We strongly favor the former possibility, because it also helps to explain the observed decrease in run length under assisting load, as well as the force dependence of exponential fits to the run length data (*Figure 3* and *Table 1*). Put another way, if load-dependent rate $k_3$ was instead to correspond to tethered-head binding, then an increase in $k_3$ under assisting load would increase the likelihood of completing a step, and thereby increase run lengths in this force regime, contrary to observation (*Figure 3*). With these assignments, $k_2$ remains the sole force-independent rate, corresponding primarily to biochemical, and not mechanical, events.

## Kinesin inter-head tension preferentially enhances the unbinding rate of the front head relative to that of the rear head

In the 2-HB state [C], inter-head tension exerts not only a forward load on the rear head, but also imposes a matching rearward load on the front head (*Figure 1B*). Having determined values for the inter-head tension and the force-dependent rate of rear-head release (*Table 2*), it is also necessary to quantify the force-dependent rate of front-head release in order to model how inter-head tension affects the unbinding gate. The application of hindering loads to the kinesin stalk places a rearward load on its front head, and therefore can serve as a proxy for estimating the rate of detachment of the front head at the unbinding gate. Previously, we showed that large superstall forces imposed by an optical trap (exceeding −7 pN) slow down the forward stepping pathway to such an extent that unbinding proceeds through a competing (slow) ATP-dependent pathway (*Clancy et al., 2011*). To quantify the rate of front-head detachment, we examined the unbinding of DmK-WT from the MT under load in the presence of saturating levels of ATP (2 mM). Measurements of unbinding rates

under load (*Figure 6*) were fit to the exponential function, $k_{off} = k_{off}^0 \exp[|F_{trap}|\delta_{off}/k_B T]$ where $k_{off}^0$ is the unloaded rate and $\delta_{off}$ is the associated distance parameter. The unbinding rate for all hindering loads ($k_{off-}$) is determined by $k_{off-}^0 = 1.11 \pm 0.03$ s$^{-1}$ and $\delta_{off-} = 0.60 \pm 0.01$ nm (fitting the unbinding rates only to data from superstall forces yielded nearly identical parameter values). From these numbers, it follows that the 26-pN inter-head tension in WT kinesin can enhance the rate of front-head release at the unbinding gate by a factor of ~50.

For assisting loads, we can estimate the unbinding characteristics of kinesin dimers from the MT (*Figure 6*) by dividing the force-dependent velocities (*Figure 5*) by their corresponding run lengths (*Figure 3*), yielding $k_{off+}^0 = 7.4 \pm 0.5$ s$^{-1}$ and $\delta_{off+} = 0.32 \pm 0.02$ nm. Taken together, the values of $k_{off}^0$ and $\delta_{off}$ for assisting and hindering loads, which are also useful for modeling transport by multiple kinesin motors, express the asymmetry in kinesin velocity and run length with respect to the direction of load. For assisting loads, the distance parameter, $\delta_{off+}$, is controlled by the run length force dependence, and we note that its value is similar to the distance parameter derived from fits to the velocity data, $\delta_3$, suggesting that both processes are governed by the detachment of the rear head.

## Cysteine-light kinesin mutants do not reflect wild-type motility under load

The force–velocity relation for DmK-6AA (*Figure 5*), which carries the insert LQASQT in its NL, differs significantly from the corresponding relation for a human (HsK) cysteine-light (CL) construct (*Rice et al., 1999*) with the insert AEQKLT, HsK-CL-6AA (*Clancy et al., 2011*) (*Figure 7*). Specifically, DmK-6AA exhibits a substantially greater velocity—which is also less force dependent—than HsK-CL-6AA (*Figure 7A*). Whereas HsK-CL-6AA was capable of undertaking many rearward steps when subjected to hindering loads beyond the stall force (*Clancy et al., 2011*), no such processive backstepping was

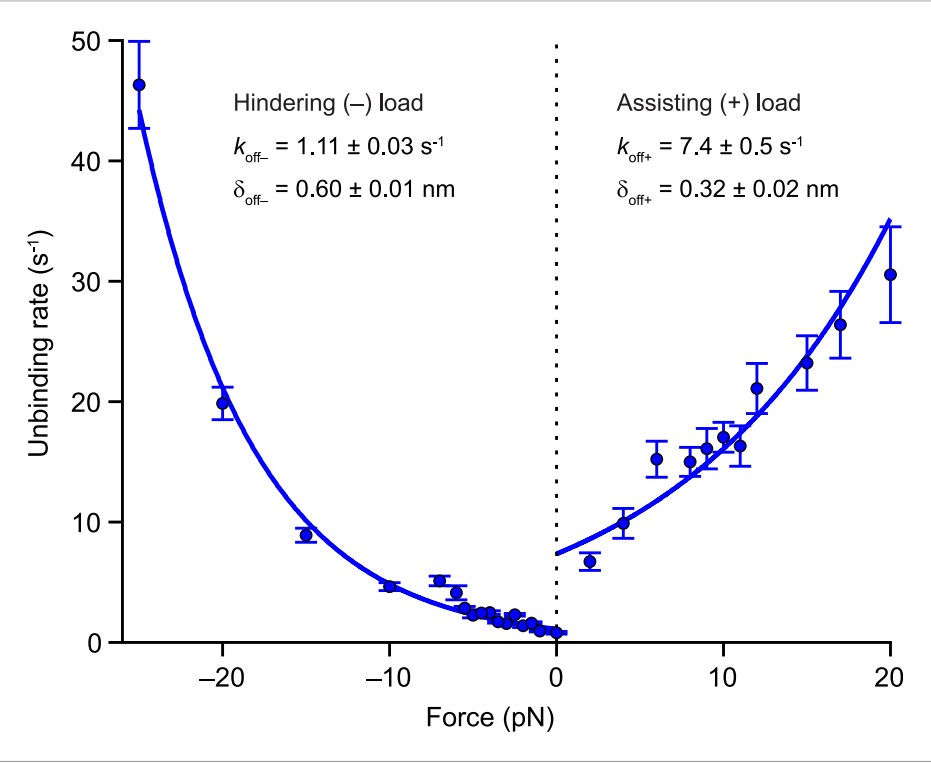

**Figure 6**. Kinesin unbinding rates are asymmetric with respect to the direction of load. Single-molecule measurements of the rate of MT unbinding for DmK-WT (mean ± SE; $N = 75–818$) at 2 mM ATP (solid circles). The unloaded release rate ($k_{off}$) and the associated distance parameter ($\delta_{off}$) were obtained from exponential fits to unbinding data acquired under hindering loads (–), –25 to 0 pN, and assisting loads (+), +2 to +20 pN. Fits (solid lines) and associated parameters (legend; mean ± SE) are shown.

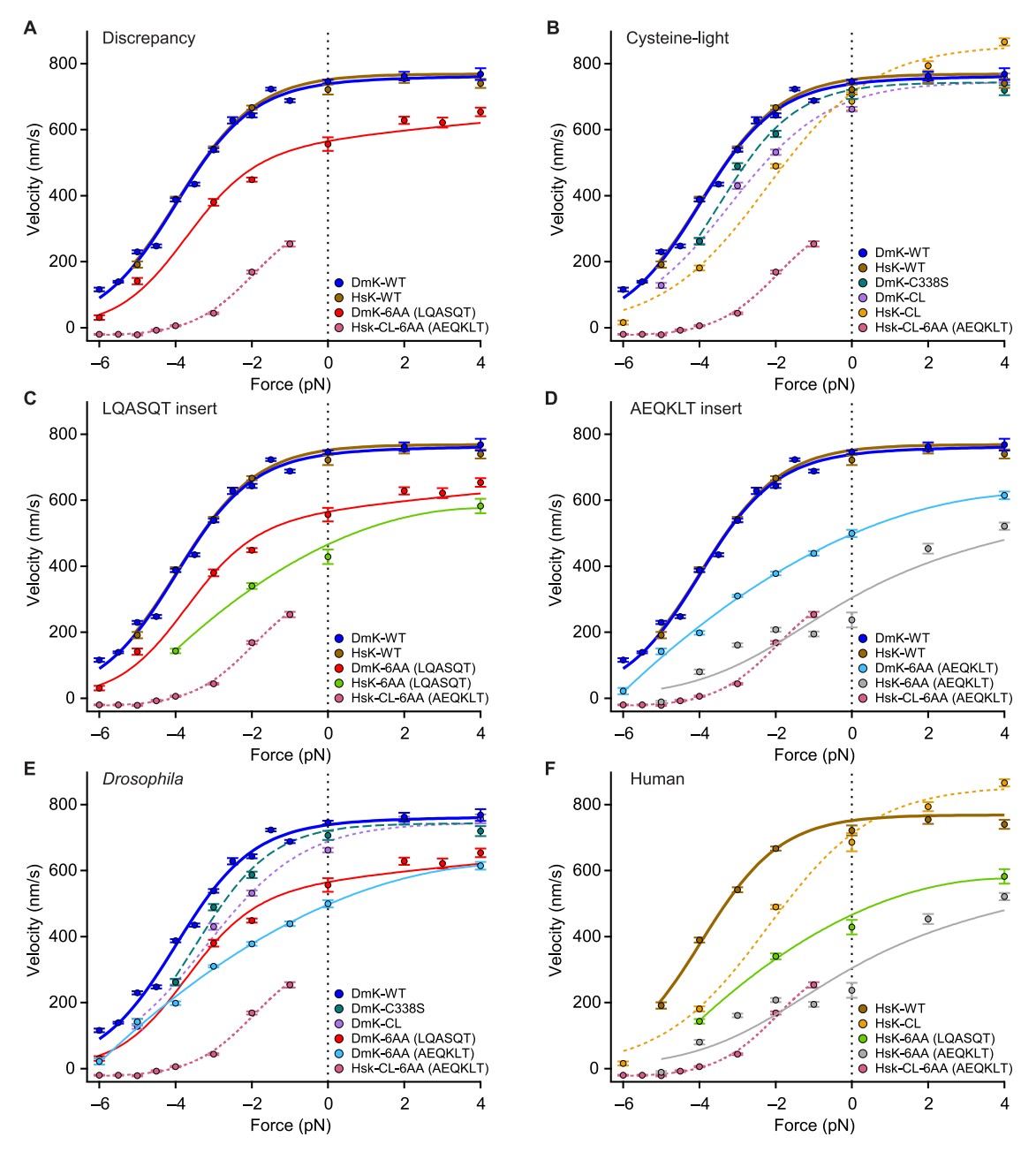

**Figure 7**. Kinesin motility characteristics depend upon parent species, NL length, NL sequence, and cysteine mutations. Force–velocity relations of constructs that differ by parent species, NL length, NL insert sequence, or cysteine mutations (mean ± SE; $N = 25$–818; solid circles, color-coded according to the legends). (**A**) DmK-6AA exhibited lower velocity under all loads than WT, but was faster and less force-dependent than HsK-CL-6AA, a human CL construct with a different 6-AA NL insert. (**B**) Side-by-side comparison of human and *Drosophila* CL mutants, along with corresponding WT constructs. Unloaded velocities of CL constructs with WT-length NL domains were similar, but CL constructs were systematically slower than WT under hindering loads. Under assisting loads, only the HsK-CL construct could be sped up beyond WT velocities. (**C** and **D**) Comparisons of constructs carrying NL insert sequences LQASQT (**C**) and AEQKLT (**D**). Force–velocity data for WT *Drosophila* and human kinesin were indistinguishable, but human constructs with 6-AA extensions of the NL moved at lower velocities than corresponding *Drosophila* motors under all loads. (**E** and **F**) Comparisons of all *Drosophila* (**E**) and human (**F**) constructs. Constructs with the NL insert AEQKLT were slower for all forces than those with LQASQT, for both *Drosophila* and human kinesin. HsK-CL-6AA data (*Clancy et al., 2011*) and hindering-load velocities for DmK-WT (*Andreasson et al., 2015*) are reproduced from previous work. Fits to DmK-WT and DmK-6AA data sets (solid lines; color-coded according to the legends) correspond to the model of *Figure 5*, with parameters values from *Table 2*. The remaining force–velocity data were fit to polynomials (solid and dashed lines; color-coded according to legends), provided to guide the eye.

observed for DmK-6AA, nor for any other *Drosophila* constructs examined that retained native cysteine residues. This finding is also consistent with the proportionally greater mantADP release by HsK-CL-6AA (*Clancy et al., 2011*) than DmK-6AA (*Figure 2A*). By contrast, the wild-type forms of *Drosophila* (DmK-WT) and human (HsK-WT) kinesin exhibit nearly identical force–velocity relations under otherwise identical conditions (*Figure 7A*). The question therefore arises whether the discrepancy between the behavior of DmK-6AA and HsK-CL-6AA is a consequence of: (i) one or more of the seven mutations used to produce the CL construct, (ii) some intrinsic difference between human and *Drosophila* kinesin, (iii) a sequence-specific effect of the NL insert, or (iv) some combination of these.

Recombinant mutant constructs have been widely used to investigate kinesin function. In particular, CL constructs have been developed to attach fluorescent labels, and it has been commonly assumed that such motors exhibit WT behavior, based on the similarity of their velocities measured under unloaded conditions (*Rice et al., 1999*). Here, we assayed the velocity of a widely used truncated human CL construct (HsK-CL) (*Rice et al., 1999*) and a *Drosophila* CL construct (DmK-CL) (*Fehr et al., 2009*) over the full range of forces (*Figure 7B*). Under hindering loads, both HsK-CL and DmK-CL were appreciably slower, and also more sensitive to load, than either DmK-WT or the corresponding truncated human construct, HsK-WT. For the case of the human CL construct, the entire shape of the force–velocity relation is affected, because this motor can be sped up under assisting loads by as much as 200 nm/s. WT motors do not exhibit this effect, and it suggests that important steps in the mechanochemical cycle have been altered in the mutant.

Both CL constructs carry, among other changes, a point mutation of a conserved cysteine in the NL domain, namely C338S in DmK-CL (*Fehr et al., 2009*) and C330S in HsK-CL (*Rice et al., 1999*). We determined the force–velocity curve for a *Drosophila* construct containing only the C338S mutation (DmK-C338S). Comparison shows that this single mutation can account for some, but not all, of the change in force dependence under hindering loads (*Figure 7B*). This indicates that a single CL mutation in the NL cannot alone account for the differences between WT and CL motors.

Although the force–velocity curves for DmK-WT and HsK-WT are practically indistinguishable, extending the NL of these motors by 6 AA, using either the sequence LQASQT or AEQKLT, showed that with the same insert sequences, human constructs were systematically slower than the corresponding *Drosophila* constructs (*Figure 7C,D*). Furthermore, NL extensions based on AEQKLT produced motors that were slower than the corresponding constructs based on LQASQT, for both DmK and HsK (*Figure 7E,F*). These findings indicate that the sequences of the inserts play some role in determining motor velocity. Nevertheless, regardless of species or insert sequence, none of the mutants with extended NL domains that retained their native cysteine residues exhibited either processive backstepping under superstall loads or velocities surpassing the WT under assisting loads. We note that because DmK-CL requires two mutations (*Fehr et al., 2009*), whereas HsK-CL necessitates seven (*Rice et al., 1999*)—only one of which is common between the two constructs—a more direct comparison of *Drosophila* and human CL constructs with extended NL domains is not practical.

## Discussion

### The stepping gate, which prevents rear-head rebinding in the ATP-waiting state, is not strongly dependent on neck linker length

The extent of mantADP release by the rear head while kinesin is in the ATP-waiting state [A] sheds light on the role of NL length in inhibiting rear-head rebinding, an essential feature of the stepping gate. Because WT and mutant constructs with as many as three additional AA in the NL can maintain a stable 1-HB ATP-waiting state, our results show that the stepping gate is not strongly influenced by NL length. Although the stepping gate is clearly compromised in constructs with longer NL inserts (4–6 AA), a substantial portion of the population retains mantADP in its rear head (*Figure 2A*), suggesting that gating is not abolished even in these motors. This notion is reinforced by results showing that mantADP exchange rates for the extended-NL mutants (*Figure 2B*) are several orders of magnitude lower than MT-stimulated ADP release rates by WT monomers, or ATP-stimulated ADP release rates by MT-bound WT dimers (*Hackney, 2002*, *2005*; *Hackney et al., 2003*). The results are fully consistent with the high degree of unidirectionality observed for all constructs, even when subjected to high hindering loads.

## Neck linker length is optimized for processivity under hindering loads or unloaded conditions

As reported previously for WT kinesin (*Milic et al., 2014*), the run lengths of constructs with extended NL domains are highly asymmetric with respect to the direction of applied load (*Figure 3*). Evidently, the binding gate is substantially more effective at maintaining processivity under unloaded or hindering-force conditions than under assisting-load conditions. Assisting loads as low as 2 pN produced a dramatic reduction in the run length, suggesting that any mechanism responsible for the binding gate becomes ineffective when kinesin is subjected to forces aligned with its overall motion. We note that models of transport by multiple kinesin motors have not yet taken load asymmetry into account (*Klumpp and Lipowsky, 2005*; *Muller et al., 2008*), but it may be important to do so explicitly, given the magnitude of the effect.

Here, we found that lengthening the NL by a single AA significantly decreased the unloaded run length, whereas further NL extensions led to negligible additional reductions (*Figure 3*). Furthermore, previous work has shown that a kinesin construct with a NL shortened by a single AA becomes non-processive (*Shastry and Hancock, 2010*, *2011*). We conclude that the length of the wild-type NL domain may be near optimal for maximizing motor processivity.

We speculate that the decrease in run length associated with extended NLs (*Figure 3*) arises from the corresponding increase in the diffusive space that the tethered head must explore before reaching its next MT binding site. Enlarging that space extends the time that the tethered head takes to reach the forward site, thereby increasing the probability that the partner head may release the MT prematurely, enhancing dimer dissociation (*Figure 1B*). This proposal is also consistent with the effect of added $P_i$ in increasing the processivity of DmK-WT and DmK-6AA (*Figure 4*): $P_i$ acts to stabilize the bound head, thereby reducing dimer dissociation. The finding that normalized run lengths for the NL constructs were identical under all conditions tested (*Figure 4B*) suggests that the rates of any biochemical events associated with the bound head, when the kinesin cycle is at the binding gate, are independent of the NL length. The absolute run lengths measured for DmK-6AA were systematically lower than those of the WT motor (*Figure 4A*), as anticipated from the increased time required for the tethered head to reach the forward MT binding site.

## Inter-head tension maximizes kinesin velocity

Extending the NL by a one AA reduced kinesin velocity under all loads, whereas additional NL extensions, up to 6 AA, produced force–velocity curves that were indistinguishable from DmK-1AA (*Figure 5*). We surmise that inserting a single AA into the NL introduces sufficient slack into the inter-head linkage that any further extensions become superfluous. A relatively high inter-head tension (26 ± 3 pN; *Table 2*) may be advantageous in tuning WT kinesin for increased velocity. However, because constructs with up to 6 AA inserted in the NL retained significant functionality (*Figure 5*), we infer that inter-head tension plays only a modulatory role in kinesin motility, a conclusion that gains support from the modeled force–velocity relations (*Figure 5*). The fit parameters indicate that WT levels of inter-head tension can amplify the rate of rear-head release by an order of magnitude beyond its unloaded rate. However, because this unloaded rate is already more than double the rate of the hydrolysis-associated events of the kinesin cycle, $k_2$, the rate-determining step of the full cycle cannot be rear-head release. We find instead that the 2-HB state of kinesin is primarily *front-head gated*, in the sense that the kinetic cycle of the front head is slowed while the rear head remains MT-bound. Put another way, the gating mechanism responsible for processivity in the 2-HB state does not require inter-head tension, but rather occurs through a modulation of the rates of biochemical processes occurring at the front head, including ATP hydrolysis, and possibly productive ATP binding (*Clancy et al., 2011*).

## Inter-head tension impairs, rather than aids, the unbinding gate in kinesin

Although an inter-head tension of 26 pN can enhance rear-head release 10-fold, unbinding measurements performed under hindering loads (*Figure 6*)—which serve as a proxy for front-head detachment rates at the unbinding gate [C]—reveal that such an inter-head tension would be expected to enhance front-head release 50-fold. Despite the front-head release rate being two orders of magnitude lower than the rear-head release rate in the absence of inter-head tension, the fact that

the load-sensitivity for detachment by the front head (captured by the distance parameter, $\delta_{off-} = 0.60 \pm 0.01$ nm; *Figure 6*) is nearly double that of the rear head ($\delta_3 = 0.35 \pm 0.02$ nm; *Table 2*) implies that increases in inter-head tension preferentially promote the unbinding of the front head. Kinesin is, in this sense, *negatively* rear-head gated at the 2-HB state: if anything, the presence of inter-head tension serves to undermine any contribution of the unbinding gate in maintaining processive motility.

Because inter-head tension adversely affects processivity from the 2-HB state, and because the rate of rear-head release is not rate limiting for the kinesin cycle even in the absence of inter-head tension, we conclude that it is unlikely to function as the primary mechanism for head coordination in the kinesin cycle. This conclusion seemingly runs contrary to previous publications, implicating inter-head tension as the preferred gating mechanism (*Rosenfeld et al., 2003*; *Guydosh and Block, 2006*, *2009*; *Toprak et al., 2009*; *Shastry and Hancock, 2011*). However, as we have previously argued (*Clancy et al., 2011*), results from the earlier studies can be reconciled with our own by invoking a front-head gating mechanism for kinesin where it is the *spatial orientation* of the NL (i.e., pointing forward in the docked state and rearward when undocked), rather than inter-head tension, per se, that facilitates coordination of biochemical states, as discussed (*Asenjo et al., 2003*; *Hahlen et al., 2006*; *Clancy et al., 2011*). In WT kinesin, an essential difference between the 2-HB state [C] and the 1-HB state with the tethered head positioned in front of the bound head [B] (*Figure 1B*) is that inter-head tension can develop only when both heads are strongly bound. Although tension may accelerate the rear-head release, no increase in this rate is necessary for gating, so long as the biochemical events in the front head of a 2-HB motor cannot proceed. In other words, gating at the 2-HB state may be accomplished not by accelerating rates associated with the rear head, but by inhibiting—via the spatial orientation of the NL—the ATP hydrolysis cycle at the front head, until the rear head detaches.

## Motility properties of kinesin constructs with extended neck linkers depend on species, insert sequence, and cysteine-light mutations

Evidently, the species of origin, the mutation of cysteines in the catalytic domain, as well as the sequence and length of the NL, can individually affect kinesin motility (*Figure 7A–F*). Although the ability of certain mutants to backstep processively under superstall loads was previously attributed to an increase in NL length (*Yildiz et al., 2008*; *Clancy et al., 2011*), the dramatic difference between DmK-6AA and HsK-CL-6AA shows that extension of the NL alone cannot account for the behavior of HsK-CL-6AA (*Clancy et al., 2011*) and related constructs (*Yildiz et al., 2008*). The AEQKLT insert produced similar velocities for HsK-CL-6AA and the non-CL version of the human construct, HsK-6AA (AEQKLT) (*Figure 7C–F*), but only the CL version could backstep processively. Based on these findings, we conclude that the processive backstepping behavior of HsK-CL-6AA arises from some combination of NL extension and the introduction of one or more of the seven mutations used to produce human CL kinesin. Further work will be required to understand the detailed basis of any such effects.

Although we stress that our specific findings on kinesin gating, together with the general gating framework presented here, are not based on data from CL mutants, numerous publications have made extensive use of these (*Rice et al., 1999*; *Case et al., 2000*; *Tomishige and Vale, 2000*; *Peterman et al., 2001*; *Rosenfeld et al., 2001*, *2002*; *Sosa et al., 2001*; *Sindelar et al., 2002*; *Asenjo et al., 2003*; *Rice et al., 2003*; *Rosenfeld et al., 2003*; *Naber et al., 2003a*, *2003b*; *Yildiz et al., 2004*; *Asenjo et al., 2006*; *Milescu et al., 2006*; *Tomishige et al., 2006*; *Mori et al., 2007*; *Sindelar and Downing, 2007*, *2010*; *Verbrugge et al., 2007*; *Dietrich et al., 2008*; *Yildiz et al., 2008*; *Asenjo and Sosa, 2009*; *Toprak et al., 2009*; *Wong et al., 2009*; *Verbrugge et al., 2009a*, *2009b*, *2009c*; *Clancy et al., 2011*; *Naber et al., 2011*; *Mattson-Hoss et al., 2014*). In light of the unusual behavior displayed by certain CL mutants, it remains to be determined which previous findings are applicable to kinesin in its WT form. To avoid artifacts in future work, it would be prudent to explore alternative linking chemistries for site-specific labels that do not require the elimination of native cysteine residues. Furthermore, wild-type behavior for mutant constructs should no longer be assumed simply on the basis of their unloaded velocities or ATP hydrolysis rates.

The force-dependent velocities for the various *Drosophila* constructs with NL extensions (DmK-1AA through DmK-6AA) were apparently independent of the sequence used to extend the NL (*Figure 5*). However, different 6-AA NL insert sequences (LQASQT vs AEQKLT) exhibited different motile properties in both *Drosophila* and human constructs (*Figure 7C–F*). The reason for these

differences is unclear, but we conjecture that it may be due to the presence of the positively charged lysine in the AEQKLT insert. In all constructs, NL extensions were introduced at the junction of NL and the stalk to minimize any disruption to the interaction between the remaining, native NL sequence and the catalytic head domain. It is conceivable that insertions at this position nevertheless affect kinesin in some ways beyond the reduction in tension arising from increased NL length, but we consider that possibility less likely, because the data in *Figures 2–5* can be well modeled by mechanical effects. We also favor the explanation that extensions of the NL lead to negligible inter-head strain in the 2-HB state, because extensions from 1 to 6 AA increase the NL length by over 30%, yet generate no corresponding change in velocity (*Figure 5*). The gating framework (*Figure 1B*) and associated conclusions would remain intact even in the presence of residual inter-head tension in mutants, however. In that case, the value for inter-head tension in the WT ($F_{i,wt}$; *Table 2*) would be interpreted as the increase in tension (relative to the mutants) produced by its shorter NL.

## Neck linker length in wild-type kinesin has been optimized to maximize both unidirectional processivity and velocity

The general framework (*Figure 1B*) provides a structure for evaluating the relative contributions of individual gates to the kinesin cycle. Because the rate of unbinding of the front head increases faster than that of the read head under increasing inter-head tension, the unbinding gate functions most efficiently at lower tension, or in its absence. Extensions of the NL abolished inter-head tension in kinesin, yet motors remained active, albeit at reduced speeds, and their stepping gates remained fully functional, suppressing rear-head rebinding even in constructs with NLs lengthened up to three AA. These considerations, taken on their own, suggest that NL extensions from 1 to 3 AA might lead to more effective gating, and consequently to greater processivity. Instead, the most pronounced effect of NL extension was a dramatic decrease in processivity. An explanation is that the dominant effect of lengthening the NL is to compromise the binding gate, and that the deleterious effect of NL length at this gate outweighs any compensating effects of reduced inter-head tension at the unbinding gate. We conclude that the key role played by NL length in kinesin gating is twofold: (1) to enhance processivity by facilitating binding of the tethered head to the forward MT site at the binding gate, and (2) to increase velocity by accelerating rear-head release from the 2-HB state at the unbinding gate.

In general, the kinesin-1 nanomechanical properties appear to be selected for processivity and velocity. Other cytoskeletal motors, including members of the kinesin, myosin, and dynein superfamilies, may be optimized to emphasize different combinations of useful characteristics, including stall force, load-bearing ability, velocity, processivity, directionality, or the ability to work cooperatively in groups, depending upon their roles in cells. Using the general gating framework (*Figure 1B*), the strategies adopted to optimize for a specific functionality could be characterized in terms of the extent to which each gate contributes to the overall cycle of a given motor. Because gating considerations unify the coupling of biochemical states and mechanical events, these may be of value in framing future investigations, as well as in re-evaluating previous work, on how dimeric motors are coordinated (*Block, 2007*; *Gennerich and Vale, 2009*; *Kull and Endow, 2013*).

# Materials and methods

## Kinesin expression and purification

All *Drosophila melanogaster* recombinant constructs, with the exception of CL mutants, were derived from a truncated sequence, comprising the first 559 AA of the *Drosophila* kinesin-1 heavy chain (KHC) with eGFP and a 6xHis-tag engineered at the C-terminus, hereafter referred to as DmK-WT (*Shastry and Hancock, 2010*; *Milic et al., 2014*). One or more rounds of site-directed mutagenesis (QuikChange, Agilent Technologies, Santa Clara, CA) were applied to produce a series of constructs with lengthened NL domains, extended by 1 (L; DmK-1AA), 2 (HV; DmK-2AA), 3 (DAL; DmK-3AA), 4 (LAST; DmK-4AA), 5 (LASQT; DmK-5AA), or 6 AA (LQASQT; DmK-6AA). A version of DmK-6AA with the NL insert AEQKLT was also generated. These insertions were all introduced at the junction of the NL and the coiled-coil stalk, between residues T344 and A345 (*Shastry and Hancock, 2010*). The procedures for the expression of proteins in *Escherichia coli* and subsequent purification via nickel column chromatography were described previously (*Uppalapati et al., 2009*; *Shastry and Hancock, 2010*). A C338S point mutant (DmK-C338S) and a CL (DmK-CL) version with 2 mutations (C45S and

C338S) were generated from a *Drosophila* KHC truncated at residue 401, with a C-terminal 6xHis-tag (*Fehr et al., 2009*). These mutants were expressed and purified as described (*Fehr et al., 2009*). A round of site-directed mutagenesis was used to produce DmK-C338S, based on the DmK-CL construct. Human non-CL kinesin-1 motors were based on a truncated construct (HsK-WT) containing the first 595 residues of KIF5B (*Navone et al., 1992*), with a C-terminal 6xHis-tag. The expression plasmid for HsK-WT (pAF4) was created by replacing the sequence coding for a truncated *Drosophila* kinesin-1 in plasmid pCA1 (*Asbury et al., 2003*) with one corresponding to the first 595 residues of KIF5B, via cassette mutagenesis. Multiple rounds of site-directed mutagenesis were used to generate two constructs with 6 AA inserted into the NL region (LQASQT and AEQKLT; HsK-6AA). The inserted residues were positioned at the junction of the NL and common coiled-coil stalk, between residues T336 and A337, respectively. The non-CL human constructs were expressed in *E. coli* and purified on a nickel column, as described (*Fehr et al., 2009*). A truncated CL human kinesin-1 construct (HsK-CL) containing the first 560 residues of the motor domain and 7 mutations (C7S, C65A, C168A, C174S, C294A, C330S, C421A) (*Rice et al., 1999*; *Rosenfeld et al., 2002*; *Clancy et al., 2011*) was also used in our motility experiments (a gift of S Rosenfeld, Cleveland Clinic).

## Bulk fluorescence assays

Stopped-flow fluorescence experiments were performed in BRB80 buffer containing 1 mM $MgCl_2$. Measurements were made on an Applied Physics SX20 spectrofluorimeter equipped with a 356 nm excitation filter and an HQ480SP emission filter, as previously described (*Chen et al., 2015*).

## Half-site mantADP release measurements

The fraction of mantADP release from kinesin dimers upon MT binding was computed from the amplitudes of fluorescence signal decreases for MT binding ($A_{MT}$) and sequential release ($A_{SR}$) experiments. For MT binding experiments, a 30 µl solution of 0.05 µM dimeric kinesin, pre-incubated with 0.5 µM mantADP, was mixed via stopped-flow with an equivalent volume of 4 µM MTs and 10 µM taxol. The decrease in the fluorescence signal after mixing reflects the amount of mantADP released. The total amount of mantADP available for release was determined from sequential release experiments, where a 30 µl solution of 0.05 µM dimeric kinesin, pre-incubated with 0.5 µM mantADP, was mixed with an equivalent volume of 4 µM MTs, 10 µM taxol, and 2 mM ATP. The amplitudes for each fluorescence decay measurement were determined by the sum of the two amplitudes obtained from a fit of the sum of two exponentials. The fraction of mantADP release upon MT binding was computed as $A_{MT}/A_{SR}$.

## MantADP exchange measurements

MantADP·kinesin·MT complexes were generated by pre-incubating 0.2 µM dimeric kinesin with 0.5 µM mantADP, 4 µM MTs, and 10 µM taxol. Fluorescence changes were recorded following the introduction of 30 µl of BRB80 buffer to an equivalent volume of the mantADP·kinesin·MT complex via stopped-flow. Records were fit to single exponentials, from which the rate of mantADP exchange was obtained.

## Optical trapping assay

The single-molecule motility assay used in this study has been described (*Milic et al., 2014*). Motility buffers consisted of BRB80 (80 mM Pipes, 1 mM EGTA, 4 mM $MgCl_2$) at pH 6.9, with 2 mM DTT, 10 µM Taxol (Paclitaxel), and 2 mg·ml$^{-1}$ BSA. An oxygen scavenging system with final concentrations of 50 µg·ml$^{-1}$ glucose oxidase, 12 µg·ml$^{-1}$ catalase, and 1 mg·ml$^{-1}$ glucose was added to motility buffers before introduction into flow cells. An optical force clamp was implemented to acquire run length and velocity data under controlled, external loads (*Valentine et al., 2008*; *Clancy et al., 2011*). Data for DmK-5AA and all constructs with 6-AA NL inserts were collected using a force clamp with an improved detection scheme (*Milic et al., 2014*). Unloaded run length and velocity data were acquired by video tracking (*Clancy et al., 2011*). Unbinding rates under assisting and moderate hindering loads, ranging from −6 to +20 pN, were obtained directly from velocity and run length records. Unbinding rates under large hindering loads, ranging from −25 to −7 pN, were estimated separately, based on the average dwell time of the last step prior to dissociation (see below).

## Run-length measurements

The starting and ending points for each single-molecule record were identified by inspection. For DmK with 1–6 AA NL inserts, the mean run length under unloaded conditions, $L$, was obtained from an exponential fit to the histogram of individual runs, where the first bin, and all bins with less than 6 counts, were excluded from fits (*Clancy et al., 2011*; *Milic et al., 2014*). As previously described (*Milic et al., 2014*), mean run lengths under hindering loads were determined based on the number of runs, $N_{1,2}$, that fell into one of two bounded intervals: $x_1 < x < x_2$, and $x > x_2$, respectively. Assuming that run lengths are exponentially distributed, the mean run lengths were computed from the relation $L = (x_2-x_1)/\ln(N_1/N_2 + 1)$, along with an estimated standard error, $\sigma_L = L\sqrt{N_1/(N_2(N_1 + N_2))}/\ln(N_1/N_2 + 1)$. For run length data collected under loads ranging from −5 to −1 pN, the lower bound was set to $x_1 = 30$ nm, and the upper bound to $x_2 = 150$ nm. Because run lengths in the presence of a −6-pN force are exceedingly short, the corresponding bounds were set to $x_1 = 15$ nm and $x_2 = 50$ nm, respectively. Runs under assisting load conditions for DmK-5AA and DmK-6AA were collected using an improved force clamp (*Milic et al., 2014*). Mean run lengths for these two constructs were obtained using the identical two-bin method, where the limits were $x_1 = 30$ nm and $x_2 = 150$ nm for data obtained under +1 to +9 pN loads, and $x_1 = 15$ nm and $x_2 = 50$ nm for loads greater than +9 pN. For DmK-1AA, DmK-2AA, and DmK-3AA, respectively, the mean run lengths under assisting-load conditions were determined by a maximum-likelihood estimator, as described (*Milic et al., 2014*).

## Velocity measurements

Velocity measurements were collected and analyzed as described (*Milic et al., 2014*).

## Unbinding measurements

For DmK-WT, unbinding rates under force-clamped conditions from −6 to +20 pN were obtained by dividing the velocity at a given force by the corresponding run length. At high hindering loads, where kinesin stalls (−25 to −7 pN), the unbinding rates were calculated by fitting an exponential distribution to histograms of the kinesin residence times on the MT. The unbinding rate, $k_{off}$, as a function of assisting or hindering load (*Figure 6*), was fit to the function $k_{off} = k_{off}^0 \exp[|F_{trap}|\delta_{off}/k_B T]$ where $k_{off}^0$ is the unloaded release rate, $\delta_{off}$ is the characteristic distance parameter for unbinding, $F_{trap}$ is the force applied by the optical trap, and $k_B T$ is Boltzmann's constant times the absolute temperature.

## Modeling

As previously (*Clancy et al., 2011*), we implemented a formalism (*Chemla et al., 2008*) to derive an analytical expression (using Mathematica 8, Wolfram Research) for velocity, $v$, as a function of load, $F_{trap}$, based on the 3-state model with 2 force-dependent transitions (*inset*, *Figure 5*):

$$v(F_{trap}) = \frac{d_{step}k_1^0 k_2^0 k_3^0 e^{\frac{F_{trap}\delta_1 + (F_{trap}+F_i)\delta_3}{k_B T}}}{k_1^0 k_2^0 e^{\frac{F_{trap}\delta_1}{k_B T}} + k_3^0 e^{\frac{(F_{trap}+F_i)\delta_3}{k_B T}}\left(k_1^0 e^{\frac{F_{trap}\delta_1}{k_B T}} + k_2^0\right)}.$$

Here, $d_{step}$ is the kinesin step size (fixed at 8.2 nm), $F_i$ is inter-head tension, $k_B T$ is Boltzmann's constant times the absolute temperature, $k_n^0$ are the unloaded rates of the 3-state model (*Figure 5*, *inset*), and $\delta_n$ are the corresponding characteristic distance parameters for these rates. The seven free parameters (*Table 2*) were determined by a global fit to 84 velocity data points (*Figure 5*) as previously described (*Clancy et al., 2011*), using Igor Pro 6 (Wavemetrics). $F_i$ for mutant constructs ($F_{i,mutant}$) was set to 0 pN (see text).

## Acknowledgements

We thank B Clancy, C García-García, C Perez, V Schweikhard, and other members of the Block laboratory for helpful comments and discussions. We also thank S Rosenfeld (Cleveland Clinic) for generously supplying cysteine-light human kinesin constructs, A Fehr (Pacific Biosciences) for creating the expression plasmid for wild-type human kinesin, and S Shastry (University of California, Santa Cruz) and D Arginteanu (Pennsylvania State University) for preparing some of the *Drosophila* kinesin constructs used for this work. BM acknowledges the support of a Stanford Graduate Fellowship and

Bio-X Undergraduate Fellowships. This work was supported by grants to SMB (5R37GM051453) and WOH (5R01GM076476) from the National Institute of General Medical Sciences of the National Institutes of Health.

## Additional information

### Funding

| Funder | Grant reference | Author |
|---|---|---|
| National Institute of General Medical Sciences (NIGMS) | 5R37GM051453 | Steven M Block |
| National Institutes of Health (NIH) | 5R01GM076476 | William O Hancock |

The funders had no role in study design, data collection and interpretation, or the decision to submit the work for publication.

### Author contributions

JOLA, BM, Conception and design, Acquisition of data, Analysis and interpretation of data, Contributed unpublished essential data or reagents, Drafting or revising the article; G-YC, Acquisition of data, Analysis and interpretation of data; NRG, Conceived the general gating framework; WOH, Conception and design, Analysis and interpretation of data, Drafting or revising the article, Contributed unpublished essential data or reagents; SMB, Conception and design, Analysis and interpretation of data, Drafting or revising the article

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
