## [Decision Letter]

Thank you for sending your work entitled “Examining kinesin processivity within a general gating framework” for consideration at *eLife*. Your article has been favorably evaluated by Richard Losick (Senior editor) and three reviewers, one of whom is a member of our Board of Reviewing Editors.

The Reviewing editor and the other reviewers discussed their comments before we reached this decision, and the Reviewing editor has assembled the following comments to help you prepare a revised submission.

The manuscript by Andreasson et al. provides significant new insights into the molecular mechanisms underlying the highly processive motion of the microtubule-based motor protein, Kinesin-1. The presented results will likely cause a paradigm shift in the motor field as to how kinesin achieves its remarkable processivity. However, several points require additional clarifications. Please also consider the suggestions of the reviewers aimed at making the paper more clear for the general audience.

Specific comments and suggestions:

1) It would greatly help the readers who have not closely followed the field to explain in the Introduction what are the current models and controversies in the field, and which specific questions/controversies the authors are resolving in this study.

2) Interpretation of the results with the neck linker mutations: it is surprising that the different-length neck-linker mutants (which have in fact quite different AA's used as inserts) behave so similar. E.g. run lengths are basically indiscernible within experimental uncertainty (Figure 3). It is even more striking for the velocity (Figure 5). This is rather surprising, given the quite large discrepancy for the two different 6AA inserts (both for fly and human kinesin) presented in the supplement. It would be important that the authors comment on this (they are, rightly, stringent to previous work using cysteine light constructs, so for this particular aspect more explanation would be in place). In this context, is the interpretation of the effect of different insertions only in terms of neck linker length and the reduction of inter-head tension fully justified? Is it possible that the conformational changes in the neck linker induced by the insertions play a role? Is it possible that inter-head tension is maintained to some extent in all mutants, and this explains why the constructs with up to 6 AA inserted retain significant functionality? Some additional discussion would be very helpful here.

3) The authors refer only once to [78], a manuscript that describes quite similar kinesin constructs, but (mainly) using cysteine-free constructs. This last aspect is likely the reason why these data are not really discussed in detail. Surprisingly, the fluorescence polarization measurements of Asenjo et al. are mentioned and used as support for the observations / conclusions. Asenjo et al. also used a (maybe slightly different) cysteine light kinesin. It would be important to justify why certain published data are used for comparison and why other data are not.

4) Some additional explanations will be useful on the sentence “…the finding that assisting load fails to appreciably increase the WT velocity suggests that the rate of rear-head release by WT kinesin must be substantially higher than the rate-limiting step(s) of the mechanochemical cycle.” There are clarifications of this point in the Discussion, but here a bit more explanation might help.

5) In the subsection headed “The force-dependence of kinesin mutants with extended neck linkers can be accounted for by a minimal 3-state model”, the choices made for the model (which rates are force dependent, which are not, which ones are dependent on inter-head strain) could be explained in more detail. No alternative model is presented, please explain why this is the only one that makes sense and is worthy of trying. More explanation might help to get the point across.

6) The authors propose that the spatial orientation of the neck linker rather than inter-head tension facilitates the coordination of biochemical states (in the subsection entitled “Inter-head tension impairs, rather than aids, the unbinding gate in kinesin”). It would be nice to explain better what determines in this model the distinction between 2-HB state and 1-HB state with the tethered head positioned in front of the microtubule-bound head. Which signal accelerates the unbinding of the rear head upon binding of the leading head if tension is not involved and the neck linker has the same orientation in the 1-HB and 2-HB state?

7) It seems strange that results of the Supplementary file 1 are not properly incorporated into the paper. The authors present a harrowing finding, very important to the motor field: they show that commonly used cysteine-light kinesin mutants show indeed quite defective motility behavior, at least when studied in detail (force-dependent run-length, resolving individual steps). This is a very important finding that needs to be shared with the field (and should be a warning for anybody working with cysteine-light mutants!). It would be preferable if these results were properly incorporated in the Abstract and the main flow of the paper, so that they would become more visible. The authors should also explain better how these results affect their own as well as previously published conclusions.

---

## [Author Response]

*1) It would greatly help the readers who have not closely followed the field to explain in the Introduction what are the current models and controversies in the field, and which specific questions/controversies the authors are resolving in this study*.

We thank the reviewers for this suggestion, and we have revised the manuscript to improve the contextualization of our work for the non-specialist. We’ve incorporated a new paragraph at the end of the Introduction that addresses the existing models and controversies in the field, and which explains how our current study speaks to these.

*2) Interpretation of the results with the neck linker mutations: it is surprising that the different-length neck-linker mutants (which have in fact quite different AA's used as inserts) behave so similar. E.g. run lengths are basically indiscernible within experimental uncertainty (*Figure 3*). It is even more striking for the velocity (*Figure 5*). This is rather surprising, given the quite large discrepancy for the two different 6AA inserts (both for fly and human kinesin) presented in the supplement. It would be important that the authors comment on this (they are, rightly, stringent to previous work using cysteine light constructs, so for this particular aspect more explanation would be in place). In this context, is the interpretation of the effect of different insertions only in terms of neck linker length and the reduction of inter-head tension fully justified? Is it possible that the conformational changes in the neck linker induced by the insertions play a role? Is it possible that inter-head tension is maintained to some extent in all mutants, and this explains why the constructs with up to 6 AA inserted retain significant functionality? Some additional discussion would be very helpful here*.

We appreciate these observations. We, too, were rather surprised by the effects of certain NL insertions on kinesin motility. In response to the reviewers’ remarks and the concerns raised, we’ve expanded the penultimate subsection of the Discussion to expand upon the considerations above.

*3) The authors refer only once to*
[78]*, a manuscript that describes quite similar kinesin constructs, but (mainly) using cysteine-free constructs. This last aspect is likely the reason why these data are not really discussed in detail. Surprisingly, the fluorescence polarization measurements of Asenjo et al. are mentioned and used as support for the observations / conclusions. Asenjo et al. also used a (maybe slightly different) cysteine light kinesin. It would be important to justify why certain published data are used for comparison and why other data are not*.

We thank the reviewers for their comment. With all respect, we cited the [78] paper four times (and not “only once”) in our original manuscript. In the revised manuscript, we now cite it eight times. We also note that we cited three papers by Asenjo et al., one of which (Asenjo et al., Nat. Struct. Biol., 2003) is used to support the notion that kinesin is gated by the spatial orientation of the neck linker. As the reviewers correctly note, the 2008 study by Yildiz and coworkers adopted a similar strategy, probing gating through the use of kinesin constructs with extended neck linkers. A central conclusion by [78], along with related work (Yildiz et al., Science, 2004), is that the ATP-waiting state of the kinesin dimer is a two-heads-bound state (2-HB). However, at least three subsequent publications, only one of which was published by my group, have argued that this conclusion is erroneous, and that the ATP-waiting state is, in fact, a one-head-bound state (1-HB) instead (Asenjo and Sosa, PNAS, 2009; Guydosh and Block, Nature, 2009; Toprak et al*., PNAS,* 2009). The discrepancy in conclusions here might be attributable to the cysteine-light constructs employed by Yildiz et al*.*, just as the reviewers implies. However, it may equally well be due to their use of non-physiological, low-salt conditions that are known to enhance electrostatic interactions between the rear head of kinesin and the microtubule, as previously suggested by Shastry and Hancock (Curr. Biol., 2010). We recognize that the three papers that we cited from the Sosa group (Asenjo et al., Nat. Struct. Biol., 2003; Asenjo et al., Nat. Struct. Mol. Biol., 2006; Asenjo and Sosa, PNAS, 2009) also used similar constructs and buffer conditions as the fluorescence experiments of Yildiz and coworkers. However, unlike Yildiz et al*.*, the fluorescence polarization measurements carried out by Sosa et al*.* allowed them to disambiguate the 2-HB state, where the rear head is bound to the microtubule, from the 1-HB state, where the rear head may, on average, be situated near the microtubule without being bound to it.

In our opinion, any conclusions regarding the ATP-waiting state, based either on inter-head distance or the position of the tethered head, are likely to be untrustworthy from those studies that employ buffers at very low ionic strength. Clearly, kinesin molecules do not operate in cells under such extremes. Some labs have resorted to using low-*Z* buffers because these greatly increase kinesin processivity (by electrostatically preventing kinesin-microtubule dissociation), resulting in longer run lengths that are more conveniently measured. However, there is a concern that this is artifactual. We agree that understanding the salient differences among the various study conditions is important, but we feel that a detailed digression on these issues in our own manuscript would (1) require too much space and (2) detract from our main point, which is the presentation of a general framework for gating. This is a topic that would be better dealt with in a critical review of the field, where such comparisons are more appropriate.

*4) Some additional explanations will be useful on the sentence “…the finding that assisting load fails to appreciably increase the WT velocity suggests that the rate of rear-head release by WT kinesin must be substantially higher than the rate-limiting step(s) of the mechanochemical cycle.” There are clarifications of this point in the Discussion, but here a bit more explanation might help*.

We’ve accepted the reviewers’ suggestion to provide more explanation at this point, prior to the Discussion, and we therefore added a sentence at the end of the paragraph in question. That said, we would prefer not to get too far ahead of ourselves, and prefer to defer additional interpretations of the data to the Discussion section of the paper, which is the more conventional way of presenting things. Doing so allows us to discuss these particular findings in the context of the remainder of the dataset and other findings, together with a model that accounts quantitatively for the observations.

*5) In the subsection headed “The force-dependence of kinesin mutants with extended neck linkers can be accounted for by a minimal 3-state model”, the choices made for the model (which rates are force dependent, which are not, which ones are dependent on inter-head strain) could be explained in more detail. No alternative model is presented, please explain why this is the only one that makes sense and is worthy of trying. More explanation might help to get the point across*.

In our revised manuscript, we’ve significantly expanded the text at this point to explain our choices in detail. However, we note here that there is no such thing as a unique model for kinesin motor motility. Furthermore, it is not practical to model a two-head walk with any fewer than three states. Although more complex models may always be advanced that incorporate additional features and states, we contend that the three-state model represents the minimal one that’s able to capture the most important features of the data.

6) The authors propose that the spatial orientation of the neck linker rather than inter-head tension facilitates the coordination of biochemical states (in the subsection entitled “Inter-head tension impairs, rather than aids, the unbinding gate in kinesin”). It would be nice to explain better what determines in this model the distinction between 2-HB state and 1-HB state with the tethered head positioned in front of the microtubule-bound head. Which signal accelerates the unbinding of the rear head upon binding of the leading head if tension is not involved and the neck linker has the same orientation in the 1-HB and 2-HB state?

We thank the reviewers for this comment, and we’ve adjusted the manuscript accordingly. Gating is best appreciated in the context of a competition between two rates. In the 2-HB state, it’s a competition between the unbinding of the rear head vs. unbinding of the front head. In this case, although tension is certainly present in the 2-HB state for wild-type kinesin, it is the inhibition of the front head, and not the acceleration of rear-head release, that supplies the strong bias for rear-head detachment, leading to forward stepping. In the 2-HB state, it is the orientation of the NL, rather than any inter-head tension, that primarily leads to unidirectional motion along the microtubule. These are two of the key conclusions of our study, in fact. These conclusions are supported strongly by the data we present here, and they are further corroborated by evidence previously published by my group, as well as several others (which we cite). However, our conclusions do run contrary to other published findings, particularly those of the Vale and Yildiz groups (which we also cite). That said, we feel that the quality (and extent) of our data speaks for itself, and is more persuasive. Furthermore, in light of our findings about artifacts that can be produced by the use of human cys-light mutants, we feel that some of the earlier work will need to be re-examined with a closer eye (please see the next point).

*7) It seems strange that results of the Supplementary file 1 are not properly incorporated into the paper. The authors present a harrowing finding, very important to the motor field: they show that commonly used cysteine-light kinesin mutants show indeed quite defective motility behavior, at least when studied in detail (force-dependent run-length, resolving individual steps). This is a very important finding that needs to be shared with the field (and should be a warning for anybody working with cysteine-light mutants!). It would be preferable if these results were properly incorporated in the Abstract and the main flow of the paper, so that they would become more visible. The authors should also explain better how these results affect their own as well as previously published conclusions*.

We’re gratified to learn that the reviewers found our results on the unusual properties of cysteine-light mutants to be interesting and worthwhile. Yes, this study has become something of a cautionary tale for all who work with such mutants under the assumption that they exhibit wild-type behavior*.* We agree that it is necessary and important to get the word out about this. We’d initially relegated the sections on cysteine-light mutants to the Supplementary file, in order to not detract from our main story about kinesin gating. But, in response to the reviewers’ comments, we’ve now adjusted the manuscript to fully re-integrate the text previously found in Supplementary file 1 into the main paper, with some minor modifications. The previous Results section of Supplementary file 1 is now found at the end of the Results section of the main text. The previous Discussion section of Supplementary file 1 is now the penultimate subsection of Discussion in the main text (and also includes additional text to address Point 2 above). We take up the reviewers’ suggestion to explain how the cysteine-light results might affect the conclusions we’ve advanced, and include extensive references to the previous literature that used these. To that end, we’ve also added a brief paragraph towards the end of our discussion of these findings, commenting on how our results speak to existing body of kinesin literature. We’ve also modified the Abstract to mention the cysteine-light results.